# What Advice Is Currently Given to Patients with Age-Related Macular Degeneration (AMD) by Eyecare Practitioners, and How Effective Is It at Bringing about a Change in Lifestyle? A Systematic Review

**DOI:** 10.3390/nu14214652

**Published:** 2022-11-03

**Authors:** Sonali Dave, Alison Binns, Valldeflors Vinuela-Navarro, Tamsin Callaghan

**Affiliations:** 1Department of Optometry and Visual Sciences, School of Health and Psychological Sciences, University of London, Northampton Square, London EC1V 0HB, UK; 2Department of Optics and Optometry, Universitat Politècnica de Catalunya, Carrer Violinista Vellsolà, 37, 08222 Terrassa, Spain

**Keywords:** age-related macular degeneration, lifestyle, nutrition, communication, advice

## Abstract

There is currently no treatment for early/intermediate Age-related Macular Degeneration (AMD) but Eye Care Professionals (ECPs) are recommended to advise patients about modifiable lifestyle factors, including dietary changes, that can slow disease progression. The aim of this review was to understand advice currently given to patients with AMD by ECPs and to evaluate evidence regarding patient compliance. A systematic review was conducted of literature published in electronic databases: CINAHL, MEDLINE, PsycINFO, PyscARTICLES, EMBASE, AMED. Methods followed PRISMA guidelines (PROSPERO registration number: CRD42020223724). Twenty-four reports were eligible for inclusion, 12 focused on ECP experience, 7 on patient experience, and 6 on impact of advice (one paper reported on the ECP and patient experience). Studies reported that a substantial proportion of patients did not recall receiving lifestyle modification advice from their ECP (57.95%, range 2–95% across patient based studies). Practitioners were most likely to provide advice about nutritional supplements (80%, range 67–93% across ECP studies), and least likely about smoking (44%, range 28–71% across ECP studies), however supplements advised did not always comply with evidence-based guidelines. The main reason for patients not following lifestyle advice was lack of provision by the ECP (54.5%, range 21–94% across studies on the impact of advice). The review highlighted a need for more studies to understand patient preferences for receiving advice and research on ECP perceived barriers to advice provision.

## 1. Introduction 

Age related Macular Degeneration (AMD) is a progressive eye condition that leads to irreversible loss of central vision and it is the leading cause of visual impairment in developed countries [1,2,3,4,5]. The early and intermediate stages of AMD are associated with relatively modest changes in visual function, but can progress to either geographic atrophy (GA) or neovascular Age related Macular Degeneration (nAMD) [6]. Both GA and nAMD are associated with significant visual disability [7], inability to perform daily activities [8], an increased risk of depression [9,10,11], reduced well-being, mood, quality of life [12,13] and social participation [14], and increased risk of falls [15,16]. Whilst nAMD can be treated with anti-vascular endothelial growth factor drugs, there are no currently licensed treatments for early stage disease or geographic atrophy. However, observational studies have highlighted certain modifiable risk factors which may be addressed to slow the progression of the disease [17,18,19,20]. Whilst smoking is accepted to be the strongest modifiable risk factor for AMD [18,21,22], dietary changes such as increased intake of dietary xanthophylls (for example in green leafy vegetables) [23], and dietary omega 3 fatty acids and oily fish [24] and adherence to a Mediterranean style diet [20] have all been reported to help decrease the risk of AMD progression. With respect to nutritional supplements, robust data is available from the Age-Related Eye Disease Study (AREDS) and AREDS2, reporting that a formula (consisting of high dose vitamin C and E, zinc, and either beta carotene or lutein and zeaxanthin) can help to slow down AMD progression (by around 25% over 5 years) in people with intermediate AMD, or with unilateral nAMD in the fellow eye [25,26]. Although evidence regarding dietary changes is less robust than the AREDS data regarding vitamin supplementation [27], there is a general professional consensus that eating a healthy diet rich in vegetables (especially antioxidant rich green, leafy vegetables), with oily fish twice per week is likely to be beneficial and unlikely to cause harm [28].

On this basis, professional bodies advise Eye Care Practitioners (ECPs) to recommend lifestyle changes based on this evidence (smoking cessation, dietary changes and vitamin supplements where appropriate) to patients with AMD verbally and in written format and to recommend other services such as smoking cessation services to help patients make the changes. The recommendations for some professional bodies are shown in Table 1.

However, studies have demonstrated that these recommendations are not consistently followed [35,36] and not all patients recall receiving any advice [22,37]. The aim of this systematic review was to investigate what advice is currently given to patients with AMD by ECPs and how effective this advice is at motivating patients to make lifestyle changes.

## 2. Methods

The review process was consistent with PRISMA guidelines [38,39]. The following databases were searched: CINAHL, MEDLINE, PsycINFO and PyscARTICLES (via EBSCO) and EMBASE and AMED (via OVID). The search was conducted in November 2020 for studies published since 2001 using the search terms displayed in Table 2.

To be included in the review, the studies had to include people with any diagnosis of AMD and had to be an evaluation of the provision of lifestyle, smoking and nutritional advice by ECPs and/or the effectiveness of this advice in bringing about a change in behaviour.

Studies were excluded if they were not published in English language; they focused on people at risk of AMD (i.e., with no current diagnosis of AMD); the full manuscript was not available or was only a published protocol, review, letter to Editors or news article; they focused on AMD with other associated systemic and ocular conditions; they evaluated a medical treatment for AMD or advice following cataract surgery; or if they were published prior to 2001—the year of publication of the original AREDS results paper [25].

All of the records were assessed for eligibility by two authors (SD and TC) and any disagreements were resolved by consulting with the other two authors (AB and VVN). The records were organised, and duplicates were removed using Mendeley software v1.19.8 (https://www.mendeley.com accessed on 20 September 2022). The data from the included studies was extracted and recorded in a data extraction table (see Appendix A). A quality appraisal assessment was also carried out for all of the records that met the eligibility criteria using quality appraisal tools including the Joanna Briggs Institute (JBI) checklist for cross sectional surveys [40], The National Heart, Lung and Blood Institute (NHLBI) checklist for interventional audits [41] and the Critical Appraisal Skills Programme (CASP) checklists for cohort studies [42] and qualitative studies [42] the findings from these tools and a summary of the included studies are shown in Table 3. The JBI quality appraisal tools were used for the cross-sectional surveys (19/24) and case series (1/24). The CASP checklists were used for the cohort studies (2/24) and one qualitative study (1/24). There was also one interventional audit for which the NHLBI quality appraisal tool was used.

For the synthesis of the data, the descriptive-interpretive approach to the meta-analysis of qualitative data was used [63]. The review protocol was published on the PROSPERO site before commencing the literature search (PROSPERO registration number: CRD42020223724).

## 3. Results

### 3.1. Included Studies

From the searches, 1370 records were identified, and 11 records were identified from other sources such as references and background reading. Before screening the records, 448 duplicates were removed, leaving 933 records to be screened. The records were screened independently by two members of the research team (SD and TC) and 859 records were excluded. 73 reports were retrieved to be assessed for full text eligibility and 1 was not retrieved as it was an older version of a paper, already included, that had been reprinted. The 73 full texts were assessed by SD and TC. One study was taken to the other two authors (AB and FVN) who confirmed eligibility. Finally, 24 papers were included in the review. Figure 1 summarizes the distribution of the studies included in this review. Most of the studies focus on the practitioner reported experiences with one study looking at both patient and practitioner experience. (see Figure 2 for PRISMA flowchart and Table 3 for list of included studies). 

### 3.2. What Is the Patient Reported Experience of Receiving Advice from Eyecare Practitioners?

Of the 24 papers included in this review, 7 papers focused on the patient experience of lifestyle advice [22,37] and their knowledge of the risk factors of AMD [44,45,48,53,58].

Two studies which surveyed patients with AMD at a hospital clinic, both reported that a high proportion of patients had no recollection of receiving advice regarding dietary modification from their ECP [22,37]. Bott et al. (2017) surveyed 248 patients with nAMD attending a medical retina clinic in the UK regarding their recollection of lifestyle advice received and reported that, although more than half (53.1%) reported being advised to stop smoking, only 39.9% reported receiving advice regarding diet, and 24.2% recalled being recommended a nutritional supplement [22]. Shah et al. (2013) carried out a similar retrospective cross sectional telephone survey of 92 patients with AMD who had attended a single UK vitreoretinal hospital unit to investigate the patients’ recollection and understanding of lifestyle advice provided [37]. They found that 47 (51%) recalled recommendations about dietary changes, 21 (23%) about exercise, 5 (5%) about smoking cessation and 90 (98%) about AREDS-based supplements. Of those who responded, based on the advice they were given, 62% felt that making dietary changes was necessary, 76% believed that exercise and weight reduction was necessary, 74% felt the AREDS supplement was a necessity, and 80% of the people who were told about smoking cessation felt it was necessary [37]. Whilst these studies demonstrated significant gaps in the knowledge of patients, they did have limitations. For example, it was not possible to determine whether advice had been provided, and subsequently forgotten by patients, or whether the advice had not been given in the first place. Additionally, the generalisability of both studies was limited by participants being recruited from a single hospital site and were conducted in the same country, thus, the results only focus on advice provided in the UK [22,37].

### 3.3. How Much Do People with AMD Understand about the Lifestyle Risk Factors for Disease Progression?

Five studies investigated patient awareness of risk factors of AMD [44,45,48,53,58], and the source of their information. Kandula et al. (2010) and Cimarolli et al. (2012) studied patient awareness of the risk factors for AMD in the United States of America (USA) [48,53]. Kandula et al. (2010) surveyed 83 patients from a retina practice in a suburban setting [53], while Cimarolli et al. (2012) conducted telephone interviews with 99 adults who were randomly selected from an Ipsos (a market research firm in the USA) database of people with AMD [48]. Both survey-based studies reported a lack of awareness amongst AMD patients about risk factors. Cimaroli and colleagues reported that out of the 99 AMD patients surveyed, one third did not know the risk factors associated with AMD and the most common source of information for all patients was their eye care physician [48]. Similarly, in the study by Kandula and colleagues 78% of the 83 patients in the study, received their AMD information from their physician, but 89% of patients would have preferred to receive more information. Furthermore, only 21%, 48%, 37%, 48%, and 36%, of patients, respectively, correctly identified how diet, special vitamins, high blood pressure, family history, and smoking can affect AMD [53]. A strength of this study was that the random recruitment of individuals through the Ipsos database from across the country increased the external validity of the findings compared to the single site studies reported elsewhere in this report. Burgmuller et al. (2016) similarly reported that, of 271 patients with AMD visiting a hospital clinic in Germany over 9 months who were asked what factors have a positive influence on their disease, only 61.7% of patients mentioned a healthy lifestyle, 53% said vitamins, and 42% of patients confessed that their knowledge of AMD was not sufficient [44].

Stevens et al. (2014) aimed to characterise AMD patients who seek the services of the Macular Society in the UK, and to determine the level and source of their knowledge about dietary recommendations for people with AMD [58]. The Macular Society is a voluntary organisation which advocates for people with AMD, and provides services including provision of information and support [64]. Stevens et al. (2014) conducted a telephone survey of 158 Macular Society members with AMD and found that just over half (55%) of the patients felt that diet was important for their eye health. Similar to the study by Kandula et al. (2010), the majority of patients (63%) did not feel that they had received enough information about AMD. Ninety-two percent of patients in this study got their information about AMD from the Macular Society, which most likely reflects the recruitment of participants from the membership of this society. However, it is interesting to note that awareness of the impact of diet on eye health remained low even in a group of individuals sufficiently motivated to join a patient advocate and support group such as the Macular Society.

Patient understanding of the risks associated with tobacco use and the potential benefit of smoking cessation was only investigated in one study [45]. Surveys were completed by 46 ECPs and 52 patients with AMD. 54% of the patients with AMD were not certain whether smoking caused macular degeneration and 90% of the people who smoked reported never being advised to quit by their ECP.

Overall, there is good evidence from these 5 studies [44,45,48,53,58] that patients attending eye clinics in the UK, US and Germany do not receive sufficient lifestyle advice to ensure a high level of understanding of the possible risks and benefits associated with diet and smoking related factors. Given the patient reported survey design of these studies, it is not possible from this evidence to determine whether the deficit is in the provision of advice, or patient recall. However, this does indicate that advice which is provided is not necessarily in a format which facilitates ready recall. There is also evidence that a significant number of patients resort to voluntary organisations such as the Macular Society to plug gaps in their knowledge of their condition [58]. One area in which evidence was lacking was regarding patient preferences with regard to modes of advice provision. This is an area that has not been investigated for AMD patients to date.

### 3.4. What Is the Practitioner Reported Experience of Advice Provision?

Twelve studies included in this review were based on practitioner reported experiences. Out of the 12 studies, seven papers related to diet, smoking and vitamin supplement advice, three focused solely on advice about vitamin supplements and 2 focused on smoking advice.

Lawrenson and Evans (2013) surveyed 1468 UK based ECPs (1414 optometrists and 54 ophthalmologists) about the lifestyle advice currently given to patients with AMD. Sixty-eight percent of the practitioners reported that they would always or usually provide dietary advice to patients with established AMD. Although 93% of practitioners recommended nutritional supplements to patients with AMD, for the majority the vitamins recommended did not comply with best evidence-based practice for nutritional supplementation in AMD, i.e., not based on AREDS guidelines [25,26]. With regard to smoking, only 32% of practitioners reported routinely taking a smoking history from patients, and 49% of the practitioners in the study reported informing patients about the link between smoking and AMD. However, 70% of practitioners took smoking history into account when recommending supplements, indicating an awareness of the possible risks of recommending certain vitamins to patient who smoke [35].

Downie and Keller (2015) carried out an online survey of 379 optometrists in Australia and similarly found that only 47% of the optometrists reported routinely asking patients if they smoke, 62% reported counselling their patients with regard to diet and 91% of recommended nutritional supplements to patients with AMD [49]. It was not clear whether the specific supplements recommended were informed by the best evidence-based guidelines, however the main supplement recommended was a high dose antioxidant which may be compliant with the AREDS formula (depending on the dosage of the specific product recommended). This is similar to the findings of Lawrenson and Evans, with less than half of the ECP’s in both studies taking a smoking history from patients but most ECP’s recommending nutritional supplements (whether appropriately or otherwise). However, Downie and Keller did report that most (88.5%) of respondents obtained their information and evidence base from peer reviewed journals, whilst non peer reviewed articles were used by 43.4% of respondents. This is in contrast to the finding of Lawrenson et al. (2013) that only 16.4% of respondents referred to scientific/research literature, and the majority were dependent on non-peer reviewed articles in professional journals [35]. This suggests the potential of some mismatch between the sources of information employed by optometrists in different countries.

In another study evaluating only optometrists, Sahli et al. (2020) administered postal surveys to 42 optometrists to examine the lifestyle advice that optometrists offer, to whom such advice is offered and reasons for not offering advice [57]. In contrast to the previous studies described above, this study found that 74% provided advice about smoking, 81% about the importance of a healthy diet and 79% regarding dietary supplements. The number of optometrists discussing smoking with patients with AMD was substantially higher in this study compared to others, but the percentage of practitioners offering dietary supplement advice was lower than previously reported [57]. However, the sample in this study was smaller than the other studies despite participants being contacted 3 times to encourage a response. The study had an overall low response rate (31% of 142 optometrists that were contacted) so the results may not be generalisable to the rest of the population.

Downie and Keller [49] and Sahli et al. [57] only surveyed optometrists so the experience of lifestyle advice provision by ophthalmologists was not reported. This is significant as Martin (2017), looking at lifestyle advice given by optometrists (*n* = 323) and ophthalmologists (*n* = 48) in Sweden, reported that ophthalmologists were more likely to provide smoking cessation advice than optometrists [36]. Lawrenson et al. (2013) also reported a higher rate of discussion about smoking cessation in their sub-analysis of ophthalmologists (as compared to optometrists, ~70% vs. ~30%). Martin et al. (2017) reported that optometrists were more likely to provide advice about nutritional supplements and diet than ophthalmologists, and found that 75% of all of the optometrists and ophthalmologists surveyed would recommend nutritional supplements to patients with late AMD in one eye and early in the other [36]. However, Lawrenson and Evans (2013) reported that ophthalmologists were more likely than optometrists (70% vs. 26%) to offer an appropriate AREDS based formula in this situation, suggesting that the optometrists surveyed in the UK were less aware of the evidence base than their ophthalmologist counterparts. They also reported that ophthalmologists were more likely to ask about smoking history (~70%) compared to optometrists (~30%) [35]. Both studies highlighted the difference in lifestyle advice provision between optometrists and ophthalmologists, but it is worth noting that Lawrenson and Evans (2013) and Martin et al. (2017) included a larger number of optometrists than ophthalmologists in this study. However, in Europe, there are more optometrists than ophthalmologists so this may explain the difference [65]. Furthermore, as in all such studies, the sample is self-selecting, meaning that those clinicians who choose to respond may be individuals with an increased interest in the topic, ophthalmologists who have specialised in AMD and therefore have a greater motivation to keep abreast of the relevant literature.

In a larger sample specifically targeting ophthalmologists, Aslam et al. (2014) evaluated ophthalmologists’ opinion of, and use of, nutritional dietary supplements 10 years after the publication of the first Age-related Eye Disease Study (AREDS). This study surveyed 216 participants (112 general ophthalmologists and 104 retinal specialists) from 7 different European countries (Belgium, France, Germany, Italy, Portugal, Spain and UK) and found that, on average, information about the benefits of nutritional supplements was regularly given to patients with AMD by 67% of ophthalmologists (a figure comparable to the findings of both Martin and Lawrenson and Evans [35,36]). Sixty-eight percent of ophthalmologists reported most commonly initiating primary prescriptions or providing advice on nutritional supplements [43]. However, no optometrists were involved in the study, and the ophthalmologists surveyed may have been unaware of advice previously provided by other healthcare professionals. A strength of this study was that ophthalmologists were asked specifically about their provision of AREDS compliant supplements, removing any doubt about whether supplements provided were consistent with evidence-based guidelines. However, this could also be considered a limitation of this study as they did not include other variations of the AREDS supplements which may have caused this percentage to be higher.

Other studies have been more specific in the aspects of nutritional advice evaluated. For example, Larson and Cocker (2009) investigated the perceptions, recommendations and educational or informational materials of licensed Wisconsin optometrists on lutein and zeaxanthin and eye health. Although the AREDS2 findings did not support the recommendation of lutein and zeaxanthin supplements to well-nourished individuals [54], there is still evidence to suggest that a diet rich in xanthophylls is beneficial to slowing progression of AMD [24,66,67,68], and this forms part of the guidelines for patient advice of most optometric/ophthalmic bodies [28,30,69]. Of the 127 practitioners in this study, 78% felt that the information available on lutein and zeaxanthin and eye health is adequate for them to make recommendations to patients. Eighty-one point one percent reported recommending lutein and zeaxanthin to patients diagnosed with AMD and 79.5% of optometrists distributed informational materials to patients [54].

Similarly, although AREDS2 did not find a benefit to the inclusion of omega 3 supplements in the AREDS formula, there is still evidence from observational studies (adopted by most practitioner guidelines) that inclusion of dietary omega-3, for example in oily fish, is beneficial to slowing AMD progression [24,70,71]. Zhang et al. (2020) looked specifically at recommendations regarding omega-3 intake given to patients with AMD by 206 optometrists from Australia and New Zealand. Optometrists reported recommending omega-3 rich foods for AMD (68%) with 95% recommending fish or non-fish seafood as a source. However, in accordance with the lack of supporting evidence, only 29% recommended specific doses of omega-3 fatty acid supplements to patients [62].

Two studies specifically assessed provision of advice on smoking cessation by practitioners [45,55]. Caban-Martinez et al. surveyed practitioners (clinical faculty, fellows and residents) based in the United States about their experiences with providing smoking cessation recommendations to patients with AMD [45]. The 46 practitioners involved in the study were asked about their smoking cessation recommendation practices and said they asked about patients smoking status all the time (13%), periodically/seldom (80%) and never (7%). When asked if they advised patients to quit smoking, 28% said always, 65% said periodically/seldom and 7% said never. This is similar to the findings by Lawrenson and Evans (2013), Martin (2017) and Downie and Keller (2015) who reported that practitioners do not always ask about patients smoking status and history [35,36,49], but this study only included ophthalmologists in a hospital setting and no optometrists. A pilot study by Lawrenson, Roberts and Offord (2014) surveying 26 UK optometrists reported that, while 77% were aware of the link between smoking and AMD, only 4% regularly took a smoking history from patients and 12% provided advice about stopping smoking to AMD patients [55]. The most common barriers to providing smoking cessation advice was the potential effect on the practitioner-patient relationship (39%), being unsure how to raise the issue (31%) and time constraints (31%). Both studies demonstrate that practitioners are not regularly asking about smoking, despite knowing the link between smoking and AMD. The studies were also carried out in different countries, thus increasing the generalisability of the findings.

Having identified that there are limitations in the provision of lifestyle advice to people with AMD, there has been some effort to explore barriers to this advice provision. Jalbert et al. (2020) surveyed 77 eye care professionals and reported that cost/funding, patient understanding/denial, discipline silos, access/availability of services and willingness to make lifestyle changes were the most commonly reported barrier for practitioners to administer effective AMD care [52]. As a potential solution to the issue, Gocuk et al. (2020) investigated whether performing clinical self-audit and receiving analytical feedback improved clinical record documentation for patients with AMD and enhanced reported provision of advice to patients. To do this, they conducted an interventional audit on 50 eye care practitioners (20 completed the study) practicing and routinely managing patients with AMD. Practitioners audited their own records for AMD patients for 3 months and were surveyed before and after the intervention. Post audit, average record documentation improved for asking about smoking status (21% to 58%), diet (11% to 29%) and nutritional supplementation (20% to 51%). Overall, optometrists’ recording of having provided lifestyle advice improved. However, before the end of the study, 30/50 optometrists dropped out, with the main reason being due to the time commitment of having to audit records, suggesting that this may not be a sustainable intervention [50]. It is also unclear from this study whether clinicians increased the frequency of advice provision, or merely became more thorough in their record keeping.

To summarise, practitioners seem to be more likely to give advice about diet and nutrition than smoking cessation advice, possibly in part because of concerns about a negative effect on the relationship between patient and practitioner of asking questions which might be perceived as being judgmental [55,72]. Figure 3 summarises the reported proportions of optometrists and ophthalmologists giving lifestyle advice. Evidence suggested that ophthalmologists are possibly more likely than optometrists to provide advice on nutritional supplements [36], and the advice given in this respect by ophthalmologists may be more compliant with evidence based guidelines [35]. Ophthalmologists may also be more likely to give advice about smoking cessation. However, comparison between practitioners is limited on small sample sizes.

### 3.5. How Much of the Lifestyle Change Advice Is Enacted?

Six studies included in this review examined the changes that patients with AMD made to their lifestyle following the receipt of lifestyle advice from their practitioners. Shah et al. (2013) asked the 92 AMD patients surveyed in their study about their compliance to the lifestyle advice they were given [37]. Adherence to diet modification advice was 81% of 47 participants who recalled advice about diet, 76% of 21 participants who recalled advice about exercise and weight reduction, and 88% of the 90 patients who recalled advice about AREDS supplementation. This suggested that advice provided by ECPs and recalled by patients did have the ability to effect a change in dietary behaviour. However, none of the 5 people who recalled being given smoking advice adhered to the recommendation.

Weaver and Beaumont (2015) aimed to understand lifestyle changes that patients make as a result of the way advice is given [59]. They found after interviewing patients attending two different clinics (clinic 1 with a strict protocol driven regime about giving lifestyle advice and clinic 2 that had no policy), that 81.6% of patients attending clinic 1 made lifestyle changes consistent with the advice they were given compared to 44% of patients in clinic 2. However, the study did not specify what the changes were which is important as the study by Shah et al. found that compliance differed between the type of lifestyle advice given [37].

Six survey-based studies specifically studied the initiation of vitamin supplement intake and dietary changes that patients with AMD made as a result of advice received. Chang et al. (2002) surveyed 108 patients with AMD recruited from a retinal specialist clinic in Canada [46]. They found that 49/108 were using supplements specifically for their AMD (45%), although 85/108 (79%) were taking vitamin supplements for general health purposes. Of those taking nutritional supplements specifically for their eye health, 33/49 (67%) were using the supplements recommended by their ECP. Similar findings were reported in a study by Charkoudian et al. (2008) where 332 new and returning patients were recruited from the retina division in a hospital in the United States of America. Two hundred and forty one (72%) of the patients were taking any supplements and 70% of these patients were taking an AREDS compliant formula. However, they reported that many of the patients did not understand why they had to use the supplements [47]. Hochstetler et al. (2010) and Parodi et al. (2016) also both reported on the rates of adherence to vitamin supplement recommendations in patients with AMD (*n* = 64 and *n* = 193, respectively). In the Hochstetler et al. (2010) study, participants were all recruited from the retina clinic of a single retinal specialist in the USA. Fifty-nine percent of the patients reported taking a vitamin supplement for AMD, with 71% of these being AREDS based. All of the participants taking supplements were recommended to do so by their retinal specialist. Seventy-five percent of the participants who did not take supplements said this was because it was never recommended to them [51]. Parodi et al. (2016) also recruited patients from a single retinal clinic in a hospital based in Milan, Italy [56]. They reported that 40% of the patients were taking AREDS supplements and, similar to the Hochstetler et al. (2010) findings, 94% of the patients not taking supplements reported that this was because it was never recommended to them [56].

These studies [46,47,51,56] all shared the limitation of recruiting participants from a single hospital site in the same country, thus reducing the generalisability of the findings. Additionally, the severity of AMD status of the participants was not categorised in two of these studies [46,51], which is important as the AREDS trial results specifically recommended the formula for patients who have intermediate AMD or advanced AMD in the fellow eye [25].

Yu et al. (2014) also reported similar findings in a German cohort [60,61]. The first study surveyed 47 patients with AMD attending eye clinics in Germany and found that 66% were recommended oral antioxidant supplements from their referring ophthalmologist, 68.1% of the total cohort were taking oral supplements for AMD, and 21.3% had never received a recommendation for supplements [60]. The second study found that 36 out of 65 patients (55%) were taking oral anti-oxidant supplements for AMD with the most common source of recommendations being from an ophthalmologist (55.4%) and, as reported in previous studies, the main reason (69%) for not taking supplements was there being no recommendation [61].

In summary, there was minimal evidence regarding compliance of patients to advice regarding general dietary changes, with the majority of studies focusing on compliance to vitamin supplement recommendations. The proportion of patients taking vitamin supplements for AMD in the included studies varied widely between around 40% and 68% [46,47,51,56,60,61]. It was not always clear whether these supplements conformed to AREDS guidelines. It also was not always apparent whether lifestyle changes of those surveyed were made directly as a result of ECP advice, but there was evidence from several studies to suggest that advice received from ECPs was impactful, particularly advice about nutritional supplements [37,46] and that the majority of people who were not making lifestyle changes were failing to do so because ECP advice had not been provided [51,56]. There was also evidence from one study to suggest that the way in which advice is provided can have a significant impact on outcomes [59].

## 4. Discussion

Overall, the studies included in this review have highlighted significant limitations in lifestyle modification advice provided by ECPs to patients with AMD.

### 4.1. The Patient Experience

This review highlights a number of key issues related to the patient experience or receiving life-style advice. Firstly, patient awareness of the risk factors for AMD in the included studies was poor. A review by Armstrong and Mousavi (2015) discussed the reported risk factors for AMD and highlighted that factors including smoking cessation, dietary changes, and regular use of dietary supplements should all be considered to reduce the lifetime risk of AMD and that ECP’s should work to increase patient knowledge of these risk factors [73]. However, the reports in this review show that despite the majority of patients citing their ECP as their main source of AMD information, they still believe they do not have enough information. This suggests that the information may not be provided to patients or they are not able to recall it [22,37]. When advice was recalled and not acted on, patients reported that it was because they felt the change was not necessary or that they lacked understanding about how it would help, suggesting that further information about the benefit of the lifestyle change is required to enhance participant adherence to advice.

However, patient reported studies have some limitations. Firstly, patients may not want their clinician to know that they are not following advice, or may not want to make negative comments about their ECP, especially when they are surveyed in the clinics. Anonymising data may help with this, but patients may still have reservations. Secondly, there is a risk of selection bias, where participants who respond may be more motivated to take part. For example, Stevens et al. (2014) recruited patients from a voluntary sector patient support group, which may have preferentially included people who were more inclined to engage with the management of their condition [58]. Thirdly, many of the studies [22,44,45,46,47,51,53,55,56,59,60,61] in this review recruited participants from single clinics. This decreases the generalisability of the results as the patients attending one clinic in one city may have different care experiences to patients in other places around the world. Finally, patient reported studies can be limited due to the incomplete patient recall of advice [37]. Patients may not always remember the advice they were given so this would not have accurately represented advice provided by ECPs. However, this may also suggest that advice may not have been administered properly or in an effective enough way to help patient recall.

The overall experience of patients with AMD in the UK has been evaluated previously (Boxell et. al., 2017). The study compared patients’ experiences of AMD care in 1999 compared to 2013 after the publication of patient management guidelines from the Royal College of Ophthalmologists [28]. A higher proportion of patients surveyed in 2013 (*n* = 1169) reported feeling satisfied overall with their diagnostic consultation overall (76% compared to 61% in 1999) [74]. Although this study did not investigate lifestyle advice specifically, studies have demonstrated that a positive health care experience can improve patient compliance [75,76].

### 4.2. The Practitioner Experience

The studies reporting practitioner experience in providing lifestyle advice for AMD found that practitioners tended to be more confident at providing advice about diet and nutrition, especially nutritional supplementation, than regarding smoking cessation. This was suggested to be at least partially attributable to concerns about a negative patient response to questions about smoking [55,72]. Between 62–81% of ECPs reported providing advice regarding dietary change (although the upper limit of the larger studies, i.e., >*n* = 100 was 68%), while advice regarding nutritional supplements was given by between 67% and 93% (with the upper limit of larger studies, i.e., >*n* = 100 being 93%) of ECPs surveyed [36,43,49,55,57]. In other words, advice on nutritional supplements was reported as being provided more frequently than advice about diet. However, there was evidence that advice regarding nutritional supplements did not always follow the most robust evidence based guidelines [35]. There was some data to suggest that ophthalmologists might be more likely than optometrists to discuss smoking cessation [35,36], and more inclined to follow AREDS [25,26] recommendations for nutritional supplement provision [35]. However, comparison between practitioners was limited by small sample sizes.

Research in other healthcare disciplines (medical, dental and nursing professionals) indicates certain common barriers which may prevent implementation of advice regarding nutrition [77]. One factor raised (alongside the issues of insufficient time, education and resources) is that healthcare practitioners feel that dietary advice guidelines can sometimes be unhelpfully vague. This may explain the finding in this review of increased confidence in providing advice regarding nutritional supplements, which is more specific and easily actioned, than advice regarding dietary change. It also emphasises the importance of a consistent and specific approach across eyecare regarding the best evidence based approach to dietary modification advice in order to give confidence to practitioners in providing the advice as well as to patients in acting upon it.

All of the studies relating to practitioner experience were questionnaire based, self-reported studies about practitioners’ opinions and practice behaviours. It can be argued that these studies can be biased by a desire for practitioners to appear in a positive light before their peers, and may not truly represent the views or behaviours of the ECP. Another potential issue is selection bias, whereby those individuals responding to a questionnaire may be those who are more engaged with research in this field and therefore more motivated with respect to providing patient lifestyle advice. However, these limitations mean that the self-reported lack of provision of dietary advice to people with AMD by one third of ECPs surveyed is likely to be a favourable representation of the true scale of advice provision.

An important point to consider is that the studies that were reported recently (2020 and later) [50,52,57,62,78] show that there are improved rates of advice provision amongst practitioners compared to earlier studies [35,36,43,45,49,54,55]. However, this review highlights that there is still a need for further education for practitioners, specifically about the importance of smoking cessation advice. This is a key factor as the evidence regarding the increased risk of AMD onset and progression associated with smoking is irrefutable. One of the largest studies on the impact of smoking on AMD, The Blue Mountains Eye Study with 3654 patients with AMD, found a significant association between smoking and neovascular AMD (OR 3.20), geographic atrophy (OR 4.54) and early AMD (OR 1.75) compared to non-smokers [21]. There have also been a number of reviews demonstrating this link and highlighting the importance of informing patients about the risk of smoking on AMD [79,80,81]. However, despite this, the 6 studies in this review that investigated smoking cessation advice given to patients, found that smoking advice was not regularly given [35,36,37,45,49,57].

This finding is not unique to ECPs. A survey of 3167 general practitioners from four Scandinavian countries reported that, of the 67% who responded, the majority did not explicitly ask the patient about their smoking history unless they displayed smoking related symptoms, and few practitioners signposted smoking cessation services [82]. Similarly, of 149 dentists surveyed in South East England, whilst 75% recorded smoking status, only around a quarter took any kind of active role in assisting them to stop. In common with the ECPs included in this review, concern regarding negative patient response was one issue highlighted, alongside a general sense that smoking cessation advice is rarely heeded, and lack of understanding of the significance of smoking to dentistry, and organisational factors (such as limited time availability) [83]. It is clear that across healthcare disciplines work is required to improve practitioner education and patient communication surrounding smoking cessation.

It is of particular concern that practitioners included in this review were also not asking about smoking history. This is crucial not just with respect to advising on smoking cessation, but also because as there is strong evidence that beta carotene supplementation increases the risk of lung cancer in smokers [84]. This means that the original AREDS formula is not appropriate for people who smoke. The AREDS2 study group recommended giving patients lutein and zeaxanthin as a carotenoid substitute in the formula [26]. This highlights the importance of taking a smoking history from patients, even with respect to recommending the appropriate vitamin supplement.

Given the limitations in advice provided by ECPs with respect to lifestyle modification, further exploration of the barriers limiting advice provision would be valuable to identify ways in which these barriers might be addressed.

Additionally, this literature review has identified a significant limitation in the current published evidence base. The published studies do not cover the behaviour of practitioners in all countries. In fact, all of the studies which met our inclusion criteria were based in Europe, Australia and the USA, so there is a real need for research investigating the behaviour of practitioners in areas in Asia, Africa, South America.

### 4.3. How Effective Is the Advice at Changing the Lifestyles of Patients with AMD?

In this review, the majority of studies reporting on compliance related to vitamin supplements. Overall, patients were taking the supplements they were recommended, but were unsure if they would help. Previous studies have shown that, when informing patients of new medication, it is important to inform them about what the medication is, how it will help and how long they should take it for as this improves compliance [85,86]. The importance of ECP advice is highlighted by the finding in this review that the main barrier to patients taking supplements was not having them recommended [22,51,56,61].

Finally, despite the large amount of evidence showing the benefits of smoking cessation on AMD progression, with smokers having a 4-fold increased risk of progression and former smokers having a 3-fold risk [18], there was only one study that looked at adherence to smoking cessation advice and reported that none of the participants who recalled being told to stop smoking took the advice (0 out of 5 patients). The other studies in this review show that patients are not aware of the link between smoking and AMD and practitioners are not giving the advice to patients.

### 4.4. How Can Effectiveness of Advice Provision Be Improved?

There has been research into ways of improving effectiveness of advice provision to people with AMD. Stevens, Cooke and Bartlett (2018) carried out an interventional study to see if a novel educational intervention can promote healthy eating and nutritional supplementation in people with AMD [87]. The participants (*n* = 100) allocated to the intervention group (*n* = 49) were given a leaflet and prompt card containing advice on diet and supplements, whilst participants in the control group (*n* = 51) were given a leaflet created by the UK College of Optometrists. All of the participants were followed up after 2 weeks, at which time there was evidence that participants in the intervention groups showed a larger increase in confidence that changing diet could slow progression of AMD, and were also more likely to make dietary changes. However, the follow up period of this study was short, and participants were not randomly allocated to the intervention group. Another study assessed the effectiveness of a telephone delivered intervention designed for giving dietary recommendations to people with AMD [78]. Participants in the intervention group (*n* = 77) were given a 20 min phone call every month for 4 months where they would provide advice to patients, assess their diet, help them with goal setting and arranging follow up support. The participants in the control group (*n* = 78) were given general leaflets about AMD and were followed up briefly once a month. Participants were also given a follow up call 4 months after the study was completed. After the intervention, participants in the interventional group significantly improved their dietary intakes of green leafy vegetables compared to baseline, whilst the change in the control group was not statistically significant compared to baseline. Additionally, the intervention group made more overall dietary changes compared to the control group, with a significant difference being in the consumption of nuts (*p* = 0.04) [78]. Although the intervention was beneficial, the time commitment required from the ECP makes the approach challenging to instigate in routine clinical practice. However, these studies do indicate that enhanced advice provision may have an impact on compliance in this patient group.

## 5. Conclusions

In conclusion, this review shows that the lifestyle advice given to patients varies and is not consistent amongst all practitioners. Practitioners appeared to be most confident in providing advice about nutritional supplements, and least confident with respect to smoking, however nutritional supplements advised did not always comply with evidence-based guidelines. There was evidence that patients were inclined to follow advice regarding supplements provided by ECPs, and the main reason stated for not following lifestyle modification advice was that it had not been provided by the ECP and because patients were not sure if following the advice would be useful. This highlights the potential scope for ECPs to bring about a change in patient behaviour through effective advice provision. The review highlighted a need for more patient centred studies to understand the best ways of providing advice to patients as well as research regarding how to overcome the ECP perceived barriers to effective lifestyle advice provision to facilitate the translation of research to positive outcomes.

## Figures and Tables

**Figure 1 nutrients-14-04652-f001:**
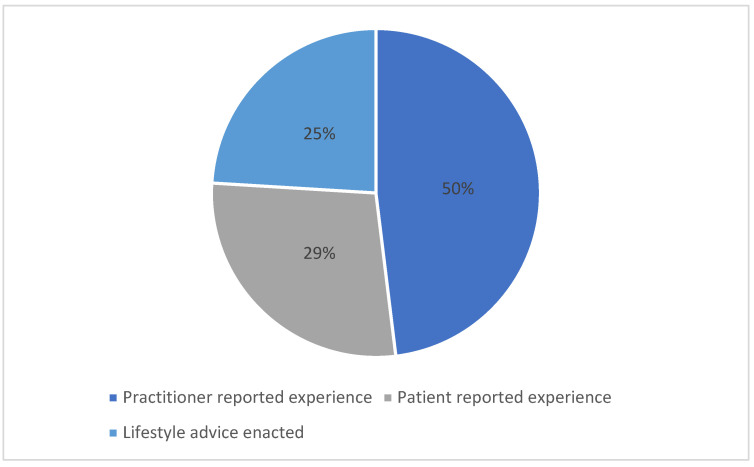
Pie chart showing how the focus of the included studies are distributed in this review. One of the papers in this review by Caban-Martinez et. al. (2011) reports on the patient experience and the practitioner experience [45]. Therefore, the percentages on this pie chart do not total 100%.

**Figure 2 nutrients-14-04652-f002:**
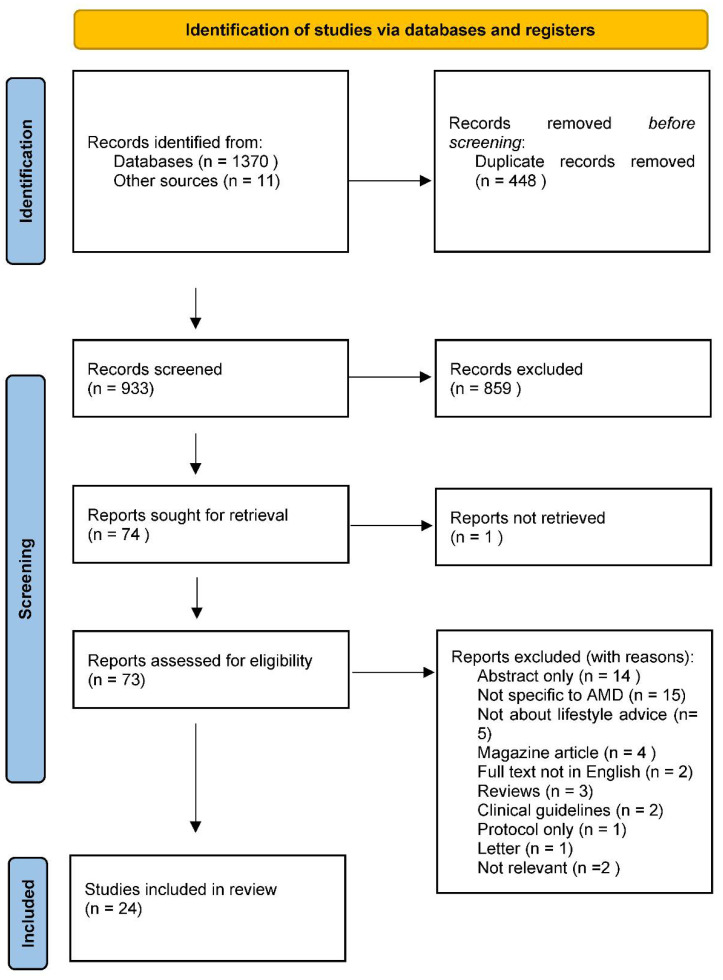
PRISMA flowchart for the selection of studies included in this review [38].

**Figure 3 nutrients-14-04652-f003:**
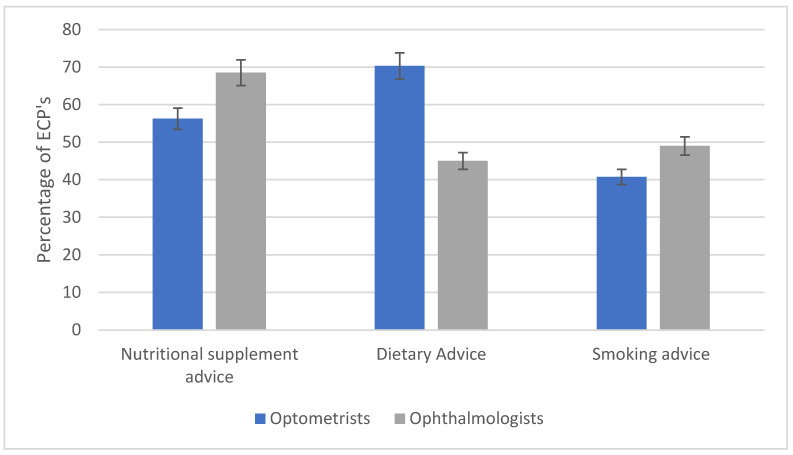
Bar chart representing the self-reported lifestyle advice given to patients about nutritional supplements, diet and smoking cessation by the two different types of ECP included in this review. Each bar represents the mean proportion of ECPs across studies who reported providing advice. Error bars denote the standard error of the mean reported value across studies. Studies contributing to these data were as follows, nutritional supplement advice optometrists [35,36,49,50,54,57], ophthalmologists [35,36,43], dietary advice optometrists [35,36,49,50,54,57,62]. ophthalmologists [35,36]. smoking advice optometrists [35,36,49,55,57] ophthalmologists [35,36,45].

**Table 1 nutrients-14-04652-t001:** Table showing a few of the professional optometry and ophthalmology associations and the lifestyle advice they are recommended to provide to patients with early AMD. *—Recommends that the RCOphth guidelines should also be followed.

Professional Body	Location	Recommendations for ECP’s
Royal College of Ophthalmologists [29]	UK	Smoking Cessation, Healthy Diet, Vitamin Supplements, Written Information
College of Optometrists * [30]	UK	Healthy Diet, Smoking Cessation, Vitamin Supplements, Written Information
American Academy of Ophthalmology [31]	USA	Smoking Cessation, Vitamin Supplements
Optometry Australia [32]	Australia	Smoking Cessation, Healthy Lifestyle
Canadian Association of Optometrists [33]	Canada	Healthy Diet, Vitamin Supplements, Sunlight protection, Smoking Cessation
International Agency for the Prevention of Blindness Africa [34]	Africa	Vitamin Supplements, Smoking Cessation

**Table 2 nutrients-14-04652-t002:** Search terms used in systematic review of electronic databases.

And	And	And	And	Not
Age-related maculopathy	Advice	Specialist	lifestyle	diabetes
age-related macular degeneration	guid *	eye care professional	diet	diabetic
age related macular degeneration	communication	eye care specialist	nutrition	genetic
macular degeneration	information	ophthalmologist	smoking	
macular disease	perception	optom *	risk factor	
	evidence based practice	clinic *	supplement	
	counselling	health care professional		
	aware *	health care provider		
	attitude *	practi *		
	behaviour	optic *		
	behavior	physician		
	recommend *	Doctor		
	experience *	Ophthalmic		
		Nurse		
		Pharmacist		

Terms within a specific column were linked with the OR operator. Terms in different columns were linked with the term in the title (And or Not). *= shortened words to widen the searches.

**Table 3 nutrients-14-04652-t003:** Table of included studies- summary of key information about the studies included in the review in alphabetical order by first author including a summary of the quality assessment of studies included in the systematic review.

Study	Location (Country and Setting)	Number of Participants	Total Study Duration	Participant Type	Study Design	Quality Appraisal Checklist Used	Risk of Bias
Aslam et al. (2014) [43]	Belgium, France, Germany, Italy, Portugal, Spain and UK	216	Not specified	Practitioners	Survey	JBI	Statistical analysis unclear, Measurement of outcome measures unclear
Bott, Huntjens and Binns (2017) [22]	UK	248	6 months	Patients	Cross sectional survey	JBI	Single site recruitment
Burgmuller et al. (2016) [44]	Germany	271	15 months	Patients	Cross sectional survey	JBI	Single site recruitment
Caban-Martinez et al. (2011) [45]	USA	98	One month	Both	Pilot cross sectional survey	JBI	Inclusion criteria not clearly defined *, Unclear if confounding factors taken into account, Measurement of outcome measures unclear, Statistical analysis unclear, Single site recruitment
Chang et al. (2002) [46]	Canada	108	2 months	Patients	Cross sectional descriptive study	JBI	Inclusion criteria not clearly defined, Statistical analysis unclear, Single site recruitment
Charkoudian et al. (2008) [47]	USA	332	2 months	Patients	Cross sectional clinical case series	JBI	Statistical analysis unclear, Single Site recruitment
Cimarolli et al. (2012) [48]	USA	99	Not specified	Patients	Descriptive study	JBI	Exposure measurement not reliable or valid, Statistical analysis unclear
Downie and Keller (2015) [49]	Australia	379	2 weeks	Practitioners	Survey	JBI	Inclusion criteria not clearly defined, Measurement of outcome measures unclear
Gocuk et al. (2020) [50]	Australia	20	17 months	Practitioners	Interventional audit	NHLBI	Sample size sufficiency unclear, Researchers not blinded to exposure
Hochstetler et al. (2010) [51]	USA	64	One month	Patients	Cross sectional survey	JBI	Inclusion Criteria not clearly defined, Single Site recruitment
Jalbert et al. (2020) [52]	Australia	77	Not specified	Practitioners	Qualitative research and focus groups	CASP	Qualitative data only
Kandula et al. (2010) [53]	USA	83	Not specified	Patients	Prospective survey based study	CASP	Unclear if confounding factors taken into account, Follow up of subjects unclear, Single Site recruitment
Larson and Coker (2009) [54]	USA	127	One month	Practitioners	Descriptive and cross sectional survey	JBI	Inclusion criteria not clearly defined, Unclear if confounding factors taken into account, Measurement of outcome measures unclear
Lawrenson and Evans (2013) [35]	UK	1468	12 weeks	Practitioners	Cross sectional survey	JBI	Inclusion criteria not clearly defined
Lawrenson, Roberts and Offord (2014) [55]	UK	26	One month	Practitioners	Survey	JBI	Inclusion Criteria not clearly defined, Exposure Measurement not reliable or valid, Unclear if confounding factors taken into account, Measurement of outcome measures unclear, Statistical analysis unclear, Single Site recruitment
Martin (2017) [36]	Sweden	393	Not specified	Practitioners	Cross sectional survey	JBI	Statistical Analysis unclear
Parodi et al. (2016) [56]	Italy	193	5 months	Patients	Cross sectional survey	JBI	Exposure measurement not reliable or valid, Single Site recruitment
Sahli et al. (2020) [57]	USA	42	Not specified	Practitioners	Survey	JBI	Unclear if confounding factors taken into account
Shah et al. (2013) [37]	UK	92	29 months	Patients	Cross sectional survey	JBI	Single site recruitment
Stevens et al. (2014) [58]	UK	158	2 months	Patients	Survey	JBI	Exposure measurement not reliable or valid
Weaver and Beaumont (2015) [59]	Australia	330	One month	Patients	Prospective controlled study	CASP	Unclear if confounding factors taken into account, Follow up of subjects unclear, Single Site recruitment
Yu et al. (2014) [60]	Germany	65	Two months	Patients	Cross sectional questionnaire based study	JBI	Single Site recruitment
Yu et al. (2014) [61]	Germany	47	Not specified	Patients	Questionnaire	JBI	Exposure Measurement not reliable or valid, Measurement of outcome measures unclear
Zhang et al. (2020) [62]	Australia and New Zealand	206	5 months	Practitioners	Survey	JBI	Inclusion Criteria not clearly defined

* = Patient questionnaire only. Quality appraisal checklists from the Joanna Briggs Institute (JBI), National Heart, Lung and Blood Institute (NHLBI) and Critical Appraisal Skills Programme (CASP) were used. The full data extraction table can be found in Appendix A and the full quality assessment checklists can be found in Appendix A.

## Data Availability

Not applicable.

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
