# Peer review of "What Advice Is Currently Given to Patients with Age-Related Macular Degeneration (AMD) by Eyecare Practitioners, and How Effective Is It at Bringing about a Change in Lifestyle? A Systematic Review"

_nutrients, 2022, doi:10.3390/nu14214652_

Round 1
Reviewer 1 Report
the manuscript is well written and designed. In my opinion, it is suitable for publication.
Author Response
Response to reviewer 1
The manuscript is well written and designed. In my opinion, it is suitable for publication.
We thank the reviewer for their positive response.
Reviewer 2 Report
“Nutrients” consider manuscripts for publication that provide novel insights into the impacts of nutrition on human health or novel methods for assessing nutritional status. Considering this, the content of this study does not seem to be in line with this aim.
In addition, this study is a summary of the 24 previously reported, but there seems to be a bias in the selected studies. For example, in Japan, many ophthalmologists ask patients with AMD about their smoking history and explain the importance of smoking cessation, and this often contributes to smoking cessation. Therefore, this is not consistent with the results of this study. Thus, the results of this study may be unapplicable in some countries.
Other points of concern include the following.
It may be difficult to understand because line numbers are not listed, but the paragraph beginning with “To summarise, practitioners seem to be more likely to give advice about diet and nutrition than smoking cessation advice,,,” before beginning with “How much of the lifestyle change advice is enacted?” contains the authors' opinions. This paragraph should be listed in the Discussion section, not in the Result section.
Author Response
Response to reviewer 2
- “Nutrients” consider manuscripts for publication that provide novel insights into the impacts of nutrition on human health or novel methods for assessing nutritional status. Considering this, the content of this study does not seem to be in line with this aim.
We thank the reviewer for their comment, but would like to present the case that this manuscript does in fact meet the stated scope of the journal and, in particular, the special issue "Age-Related Macular Degeneration: Advances in Diet and Nutrition Management”. As the reviewer highlights, Nutrients considers manuscripts for publication that provide novel insights into the impacts of nutrition on human health or novel methods for assessing nutritional status and within this public health, diet-related disorders and nutritional supplements are journal subject areas of interest. A key feature of public health research is the promotion of healthy lifestyles, in this case healthy nutrition. Furthermore, a keyword of the special issue is ‘diet intervention’ which closely relates to this work, as a dietary intervention is only effective if it is proposed to the patient at risk.
This systematic review is evaluating the evidence that clinicians are actually implementing dietary interventions informed by evidence based practice at the present time. As we develop the evidence base for new and more effective dietary interventions, it is vital that practitioner awareness and engagement advances in parallel with this improved understanding, or impact will be limited.
Finally, we would like to emphasize that the special issue "Age-Related Macular Degeneration: Advances in Diet and Nutrition Management" relates specifically to nutrition in age-related macular degeneration and is particularly interested in the evaluation and dissemination of the knowledge acquired over the past years regarding the beneficial effects of a healthy diet for AMD prevention, which this manuscript explicitly addresses.
- In addition, this study is a summary of the 24 previously reported, but there seems to be a bias in the selected studies. For example, in Japan, many ophthalmologists ask patients with AMD about their smoking history and explain the importance of smoking cessation, and this often contributes to smoking cessation. Therefore, this is not consistent with the results of this study. Thus, the results of this study may be unapplicable in some countries.
We agree with the reviewer that there is a risk of bias in any literature review process. For this reason, we conducted this systematic review according to a well-defined protocol, published and registered before the review commenced on PROSPERO (CRD42020223724). In addition, to minimize the risk of bias, two of the authors independently reviewed all potentially eligible studies against the inclusion and exclusion criteria. This review included all eligible studies.
To the authors knowledge this is a comprehensive review following a clear protocol for the selection of studies, but the authors agree that there is a limitation in the primary research addressing the focus of this systematic review. Specifically, the available published evidence does not provide a comprehensive picture of the global approach to nutritional advice in AMD. However, this in itself is an important finding.
We believe, therefore, that the issue is not with the comprehensiveness of the review, but with the limited primary published research in this field. The reviewer makes a specific point about ophthalmologists in Japan providing smoking cessation information to patients with AMD and this not having been included in the systematic review. While it is plausible that this happens in Japan, if there is no evidence of this published in international scientific peer reviewed journals, this falls outside the scope of this systematic review. If information about other countries is not presented in this systematic review is because it was not available in international peer reviewed journals or could not be accessed by the researchers due to a language barrier.
To address this issue, we have clarified in the revised manuscript that the lack of evidence in this review regarding practice in certain geographical regions reflects a deficit in the current evidence base i.e. the notable lack of evidence pertaining to clinical practice in countries outside of North America, Europe and Australia. See revised text below.
Page 19, Para 2: Additionally, this literature review has identified a significant limitation in the current published evidence base. The published studies do not cover the behaviour of practitioners in all countries. In fact, all of the studies which met our inclusion criteria were based in Europe, Australia and the USA, so there is a real need for research investigating the behaviour of practitioners in areas in Asia, Africa, South America.
- It may be difficult to understand because line numbers are not listed, but the paragraph beginning with “To summarise, practitioners seem to be more likely to give advice about diet and nutrition than smoking cessation advice,,,” before beginning with “How much of the lifestyle change advice is enacted?” contains the authors' opinions. This paragraph should be listed in the Discussion section, not in the Result section.
We thank the reviewer for bringing this point to our attention. This paragraph has now been amended and sections of this paragraph have been moved to the discussion section on page 17 paragraph 1 and page 19, paragraph 2.
Reviewer 3 Report
I read it with great interest, but I have raised several concerns.
#1. In Table 1 and 2, do not use abbreviations (VS, SC...).
#2. In Figure 1, please update PRISMA 2020 guideline form.
#3. In Method, please cite additional PRISMA 2020 guideline such as https://doi.org/10.54724/lc.2022.e9
#4. I can not see the results of risk of bias tool. This is very important issues.
#5. This paper is an excellent paper.
Author Response
Response to reviewer 3
- In Table 1 and 2, do not use abbreviations (VS, SC...).
This has been amended as requested, see Tables 1 and 3, page 2 and 7 respectively.
- In Figure 1, please update PRISMA 2020 guideline form.
This has now been amended, please see page 6.
- In Method, please cite additional PRISMA 2020 guideline such as https://doi.org/10.54724/lc.2022.e9
This additional reference has now been added on page 3, paragraph 2.
- I cannot see the results of risk of bias tool. This is very important issues.
We agree with the reviewer that the assessment of bias is a vital aspect of the systematic review. The main issues identified by the risk of bias tools were summarised in the final column of Table 3 in the original manuscript (the table of included studies). We have expanded this column of the table. In addition, we have now formatted the full results of the different risk of bias tools to include as an additional piece of supplemental material (see supplementary materials 2, 3, 4, 5 and 6).
- This paper is an excellent paper.
We thank the reviewer for their kind comment.
Reviewer 4 Report
The manuscript by Dave et al. represents a systematic review, which seeks the answers to the questions: what advice is currently given to patients with AMD by eyecare practitioners, and how effective is it at bringing about a change in lifestyle? The authors analyze the studies available but generally conclude that the information available within the literature is insufficient for making clear conclusions. Nonetheless, in my opinion the authors rise important and timely questions. I have only two minor suggestions:
- The abstract is very general. I would like to suggest authors including more particular results of the analysis performed;
- I would like to suggest the authors to think about on how to include more illustration summarizing and simplifying understanding the data presented within the Results section.
Author Response
Response to reviewer 4
We thank the reviewer for their positive feedback regarding the importance and timeliness of the questions raised by this review. In response to the specific questions raised:
- The abstract is very general. I would like to suggest authors including more particular results of the analysis performed;
We thank the reviewer for their comment. We have added more detail to the abstract, specifically the results. However, we are limited in the amount of detail we can add due to the word count restriction.
- I would like to suggest the authors to think about on how to include more illustration summarizing and simplifying understanding the data presented within the Results section.
We thank the reviewer for their comment and have now added Figures 1 and 3 on page 5 and 15 of the results section to summarize and simplify understanding of the data.
Round 2
Reviewer 2 Report
I think that the authors have amended the manuscript favorably based on the reviewers' comments.
Reviewer 3 Report
This is a mesmerizing paper.